# Lifting in Multi-agent Systems under Uncertainty

**Tanya Braun**[1]          **Marcel Gehrke**[2]          **Florian Lau**[3]          **Ralf Möller**[2]

[1]Computer Science Department, University of Münster, Münster, Germany
[2]Institute of Information Systems, University of Lübeck, Lübeck, Germany
[3]Institute of Telematics, University of Lübeck, Lübeck, Germany

## Abstract

A decentralised partially observable Markov decision problem (DecPOMDP) formalises collaborative multi-agent decision making. A solution to a DecPOMDP is a joint policy for the agents, fulfilling an optimality criterion such as maximum expected utility. A crux is that the problem is intractable regarding the number of agents. Inspired by lifted inference, this paper examines symmetries within the agent set for a potential tractability. Specifically, this paper contributes (i) specifications of counting and isomorphic symmetries, (ii) a compact encoding of symmetric DecPOMDPs as partitioned DecPOMDPs, and (iii) a formal analysis of complexity and tractability. This works allows tractability in terms of agent numbers and a new query type for isomorphic DecPOMDPs.

## 1 INTRODUCTION

Decentralised partially observable Markov decision problems (DecPOMDPs) allow for formalising an offline decision making problem for multiple agents collaborating for a joint reward in a stochastic environment. However, in the worst case, computation time in DecPOMDPs depends exponentially on the number of agents, which means that scalability is limited and crude approximations might be necessary to solve problem instances at the expense of accuracy. There are applications, though, that require accuracy. One such application lies in nanoscale medical systems in which a swarm of indistinguishable nanodevices with limited online computation power collaborate on, e.g., a treatment or diagnosis [Lau et al., 2019]. The number of agents goes into the hundreds of thousands, making the exponential dependency punishing. The question is what to do when faced with large agent sets and whether we are able to do something about the exponential dependency.

In probabilistic inference, the notion of lifting refers to efficiently handling sets of *indistinguishable* objects using representatives [Poole, 2003]. Lifting enables reducing the theoretical dependency from exponential to polynomial, making the problem of probabilistic inference tractable, w.r.t. these set sizes [Niepert and Van den Broeck, 2014]. Therefore, this paper examines indistinguishability within the agent set and its potential for making DecPOMDPs tractable regarding agent numbers, i.e., solving them no longer depends on agent numbers exponentially. The paper analyses two types of symmetries for indistinguishability inspired by the main lifting tools of counting and isomorphism in lifted variable elimination (LVE) [Taghipour, 2013]. The results rely on partitioning the set of $N$ agents into $K$ partitions with $K \ll N$, $K$ typically being in the order of magnitude 1. In nanoscale systems, e.g., the standard setting comprises a small number of agent types (e.g., $K = 4$) of which large numbers of indistinguishable agents are built.

Specifically, this paper contributes (i) an analysis of counting and isomorphic symmetries and their connection, (ii) an encoding of those symmetries as partitioned DecPOMDPs, and (iii) a formal analysis and discussion. The last part covers representation size, policy space, evaluation cost, and expressiveness. The results show that counting requires no further assumption but indistinguishability and leads to a polynomial dependence on the agent numbers except for the policy space. The isomorphic case involves a stricter symmetry, which allows for tractability in agent numbers. In addition, it enables using existing solution methods and defining new query types. Specifically, the isomorphic case allows for formalising a new optimisation problem, asking for the number of agents needed to satisfy a particular goal.

Lifting has been successfully used for query answering [Kersting et al., 2009, Van den Broeck et al., 2011, Braun and Möller, 2018, Holtzen et al., 2019] and online decision making [Nath and Domingos, 2009, Apsel and Brafman, 2011, Gehrke et al., 2019a,b], which uses decision and utility nodes in a relational or first-order probabilistic graphical model (PGM). In offline decision making, lifting has

*Accepted for the 38th Conference on Uncertainty in Artificial Intelligence* (UAI 2022).

been used in calculations for relational descriptions of the state space in (PO)MDPs: First-order MDPs (FOMPDs) [Boutilier et al., 2001] have a representation based on the situation calculus [McCarthy, 1963]. Factorised FOMDPs additionally assume a factorised representation of the state space [Sanner and Boutilier, 2007]. Sanner and Kersting [2010] use lifting for pruning indistinguishable policies in FOPOMDPs. Open-universe FOPOMDPs have an open-universe assumption about the first-order representation, using Bayesian logic as a basis [Srivastava et al., 2014]. Another first-order representation is independent choice logic that also allows for a set of agents [Poole, 1997]. To the best of our knowledge, we are the first to consider lifting for the agent set. We leave the state space in this paper as is and look at its structuring as an exciting avenue for future work. As this paper covers an analysis of the theoretical problem, we do not cover DecPOMDP solution methods as related work. Please refer to Oliehoek and Amato [2016].

This paper is structured as follows: Section 2 recaps Dec-POMDPs and their complexity results. Section 3 covers the symmetries, the compact encodings in partitioned Dec-POMDPs, and the formal analysis, followed by a discussion. Section 4 concludes the paper.

## 2 BACKGROUND AND NOTATION

This section recaps DecPOMDPs. It is set to be self-contained, but familiarity with POMDPs helps.

### 2.1 DECENTRALISED POMDPS

A DecPOMDP refers to a set of agents working jointly towards a common goal. The underlying principle is that of *maximum expected utility* (MEU) in which a utility function represents preferences over states. We look through a PGM lens for definitions, which are based on the work by Oliehoek and Amato [2016] and Russell and Norvig [2020], using random variables, $V$, which can take discrete values, referred to as range, $ran(V) = \{v_1, \ldots, v_m\}$. So-called *decision random variables* have actions as ranges. Setting $V$ to a value $v \in ran(V)$ (an *event*) is denoted as $V = v$ or $v$ for short if $V$ is clear from its context. We denote sequences over a discrete time interval $[t_s, t_e]$ with subscript $t_s{:}t_e$, e.g., $(V_{t_s}, \ldots, V_{t_e}) = V_{t_s{:}t_e}$. We use $\circ$ to denote concatenation.

**Definition 1.** *A* model $M$ *is a tuple* $(\boldsymbol{I}, S, \boldsymbol{A}, T, R, \boldsymbol{O}, \Omega)$,

- $\boldsymbol{I}$ *a set of* $N$ *agents,*
- $S$ *a random variable with a set of states as range,*
- $\boldsymbol{A} = \{A_i\}_{i \in \boldsymbol{I}}$ *a set of decision random variables* $A_i$, *each with a set of local actions as range, with* $ran(\boldsymbol{A}) = \times_{i \in \boldsymbol{I}} ran(A_i)$ *the set of* joint actions,
- $T(S', S, \boldsymbol{A}) = P(S' \mid S, \boldsymbol{A})$ *a transition function, with* $T(S_0, ., .) = P(S_0)$ *referring to a state prior,*

- $R(S, \boldsymbol{A})$ *a reward function,*
- $\boldsymbol{O} = \{O_i\}_{i \in \boldsymbol{I}}$ *a set of random variables* $O_i$, *each with a set of local observations as range, with* $ran(\boldsymbol{O}) = \times_{i \in \boldsymbol{I}} ran(O_i)$ *the set of* joint observations, *and*
- $\Omega(\boldsymbol{O}, S) = P(\boldsymbol{O} \mid S)$ *a sensor function.*

*Optional are a finite horizon* $\tau$, *a discount factor* $\gamma \in [0, 1]$ *(default 1), and an error margin* $\epsilon > 0$. *Each agent* $i \in \boldsymbol{I}$ *has a local policy* $\pi_i : ran((O_{i,0:t})) \mapsto ran(A_i)$ *mapping observation histories* $o_{i,0:t}, t \leq \tau - 1$, *to actions* $a$, *with* $\boldsymbol{\pi} = (\pi_i)_{i \in \boldsymbol{I}}$ *a joint policy. The semantics of* $M$ *is given by all possible joint policies, referred to as* $\Pi_M$. *The* DecPOMDP *asks for the joint policy* $\boldsymbol{\pi}^*$ *that yields the maximum expected utility* $U_M(\boldsymbol{\pi}^*)$ *in* $M$ *(with horizon* $\tau$*):*

$$MEU(M) = (\boldsymbol{\pi}^*, U_M(\boldsymbol{\pi}^*))$$
$$\boldsymbol{\pi}^* = \arg\max_{\boldsymbol{\pi} \in \Pi_M} U_M(\boldsymbol{\pi}) \qquad (1)$$

*where* $U_M(\boldsymbol{\pi})$ *is calculated recursively over* $\tau$ *steps and weighted according to the prior* $T(S_0, ., .)$:

$$U_M(\boldsymbol{\pi}) = \sum_{s_0 \in ran(S)} T(s_0, ., .) U_M^{\boldsymbol{\pi}}(s_0, \emptyset_{0:0}) \qquad (2)$$

$$U_M^{\boldsymbol{\pi}}(s_t, \boldsymbol{o}_{0:t}) = R(s_t, \underbrace{\boldsymbol{\pi}(\boldsymbol{o}_{0:t})}_{=\boldsymbol{a}_t}) + \gamma^t \sum_{s_{t+1} \in ran(S)} T(s_{t+1}, s_t, \boldsymbol{\pi}(\boldsymbol{o}_{0:t}))$$

$$\cdot \sum_{\boldsymbol{o}_{t+1} \in ran(\boldsymbol{O})} \Omega(\boldsymbol{o}_{t+1}, s_{t+1}) U_M^{\boldsymbol{\pi}}(s_{t+1}, \boldsymbol{o}_{0:t+1}) \qquad (3)$$

*with* $U_M^{\boldsymbol{\pi}}(s_{\tau-1}, \boldsymbol{o}_{0:\tau-1}) = \gamma^{\tau-1} R(s_{\tau-1}, \boldsymbol{\pi}(\boldsymbol{o}_{0:\tau-1}))$ *as the stopping point and* $\boldsymbol{o}_{0:t+1} = \boldsymbol{o}_{0:t} \circ \boldsymbol{o}_{t+1}$.

Each agent has its own set of actions and observations[1] whereas state and reward are joint. The joint state is usually assumed to not be fully observable, even if combining all local observations. If the joint state were observable, the DecPOMDP would simplify to a DecMDP. The joint reward function encodes that the agents receive a reward as a team. If the joint state and reward function can be split up into independent subspaces per agent, the DecPOMDP decomposes into a set of POMDPs that can be solved individually. In this paper, we do not let the problem dissolve into subproblems that can be solved independently. Rather, we keep a joint state and reward function, while the agent set shows certain degrees of independence. The generalisation of a DecPOMDP is a partially observable stochastic game (POSG) in which each agent has its own reward function $R_i$, which may conflict with other agents' rewards. The sensor function may also depend on the joint action, which does not change the problem in a major way [Russell and Norvig, 2020]. A naive solution approach is to solve Eq. (1) directly, i.e., to generate all possible joint policies, evaluate them according to Eq. (2), and pick the policy with highest value.

---

[1] $A_i = A_j$ or $O_i = O_j$, $i, j \in \boldsymbol{I}$, is possible but not mandatory.

## 2.2 COMPLEXITY RESULTS

DecPOMDPs depend exponentially on $N$, the number of agents, in the worst case, which manifests itself in the space requirements of a model $M$ of Def. 1, the size of the policy space spanned by $M$, i.e., $|\Pi_M|$, and the cost of evaluating a joint policy $\boldsymbol{\pi}$, given a horizon $\tau$. The size of the policy space together with the cost of evaluating a policy in the space provides an upper bound on any runtime as it bounds the effort of the naive solution approach to the DecPOMDP.

Formally, the sizes $\mathbb{T}$, $\mathbb{R}$, and $\mathbb{O}$ of the functions $T(S', S, \boldsymbol{A})$, $R(S, \boldsymbol{A})$, and $\Omega(\boldsymbol{O}, S)$, respectively, lie in

$$\mathbb{T} \in O\left(s^2 a^N\right) \quad \mathbb{R} \in O\left(sa^N\right) \quad \mathbb{O} \in O\left(so^N\right) \quad (4)$$

with $s = |ran(S)|$, $a = \max_{i \in \{1,\ldots,N\}} |ran(A_i)|$, and $o = \max_{i \in \{1,\ldots,N\}} |ran(O_i)|$, which are all exponential in $N$. They follow directly from the range sizes of the inputs. The cost $\mathbb{C}$ of evaluating a joint policy and the size $\mathbb{P}$ of the policy space lie in [Oliehoek and Amato, 2016]:

$$\mathbb{C} \in O\left(so^{N\tau}\right) \qquad \mathbb{P} \in O\left(a^{\frac{N(o^\tau-1)}{o-1}}\right). \quad (5)$$

Both depend on $N$ exponentially. The complexity of $\mathbb{C}$ comes from evaluating in each of the $s$ states a joint policy of size $o^{N\tau}$ where $o^\tau$ bounds the size of an agent's observation history. The complexity of $\mathbb{P}$ comes from each possible action at the end of each possible observation history, of which there are $(o^\tau - 1)/(o - 1)$ (histories form trees: geometric series).

## 3 SYMMETRIC DECPOMDPS

The idea of partitioning the agent set follows the insight that in systems with regularities in their agent set $\boldsymbol{I}$, e.g., through types, partitions emerge, in which agents have the same available actions, observations, and behaviour, i.e., they are indistinguishable. Borrowing from the field of lifted inference, we refer to these DecPOMDPs as *symmetric* since the indistinguishable behaviour materialises itself by symmetries in the transition, sensor, and reward functions. This section formalises indistinguishability through symmetries and analyses their relationship and effect on complexity. We introduce partitioned DecPOMDPs as a framework.

### 3.1 PARTITIONING OF THE AGENT SET

For lifting to apply, a DecPOMDP has to fulfil a basic requirement: The set of agents $\boldsymbol{I}$ partitions into $K$ sets $\mathfrak{I}_k$, i.e., $\boldsymbol{I} = \bigcup_{k=1}^K \mathfrak{I}_k$, $\mathfrak{I}_k \neq \emptyset$, and $\forall k, l \in \{1, \ldots, K\}, k \neq l$: $\mathfrak{I}_k \cap \mathfrak{I}_l = \emptyset$, and for each $\mathfrak{I}_k$, it holds for all pairs $i, j \in \mathfrak{I}_k$:

$$ran(A_i) = ran(A_j) \wedge ran(O_i) = ran(O_j) \quad (6)$$

Given Eq. (6), it is sufficient to keep $K$ decision and observation random variables instead of $N$ variables $A_i$ and $O_i$. To still represent the agents, one may add a logical variable $X_k$ that represents all agents in $\mathfrak{I}_k$ to $A_k$ and $O_k$, forming parameterised random variables (PRVs) $A_k(X_k)$ and $O_k(X_k)$, where the individual agents no longer appear explicitly but hidden behind the logical variable. See, e.g., the work by Taghipour [2013] for details on PRVs.

Equation (6) is a necessary and sufficient condition for using PRVs to list observations and actions available for a partition but it is only a necessary condition for lifting: Further symmetries have to hold in the transition, reward, and sensor functions for lifting to apply, of which we investigate two types inspired by the lifting tools of counting and isomorphism. Before we turn to lifting, we set up partitioned DecPOMDPs as the general framework for DecPOMDPs with partitioned agent sets fulfilling Eq. (6).

**Definition 2.** *A partitioned* model $\bar{M}$ *is a tuple* $(\bar{\boldsymbol{I}}, \bar{S}, \bar{\boldsymbol{A}}, \bar{T}, \bar{R}, \bar{\boldsymbol{O}}, \bar{\Omega})$, *with*

- $\bar{\boldsymbol{I}}$ *a partitioning* $\{\mathfrak{I}_k\}_{k=1}^K$ *of agents,* $n_k = |\mathfrak{I}_k|$ *and* $|\bar{\boldsymbol{I}}| = \sum_k n_k = N$,
- $\bar{S}$ *a random variable with a set of states as range,*
- $\bar{\boldsymbol{A}} = \{\bar{A}_k\}_{k=1}^K$ *a set of decision random variables* $\bar{A}_k$, *each with possible* partition actions *as range, with* $ran(\bar{\boldsymbol{A}}) = \times_{k=1}^K ran(\bar{A}_k)$ *the set of joint actions,*
- $\bar{T}(\bar{S}', \bar{S}, \bar{\boldsymbol{A}}) = P(\bar{S}' \mid \bar{S}, \bar{\boldsymbol{A}})$ *a transition function,*
- $\bar{R}(\bar{S}, \bar{\boldsymbol{A}})$ *a reward function,*
- $\bar{\boldsymbol{O}} = \{\bar{O}_k\}_{k=1}^K$ *a set of random variables* $\bar{O}_k$, *each with a set of* partition observations *as range, with* $ran(\bar{\boldsymbol{O}}) = \times_{k=1}^K ran(\bar{O}_k)$ *the set of joint observations, and*
- $\bar{\Omega}(\bar{\boldsymbol{O}}, \bar{S}) = P(\bar{\boldsymbol{O}} \mid \bar{S})$ *a sensor function.*

*Optional are a finite horizon* $\tau$, *a discount factor* $\gamma \in [0, 1]$ *(default 1), and an error margin* $\epsilon > 0$. *Each partition* $\mathfrak{I}_k$ *has a* partition policy $\bar{\pi}_k : ran(\bar{O}_k) \mapsto ran(\bar{A}_k)$ *mapping observation histories* $\bar{o}_{k,0:t}, t \leq \tau - 1$ *to a partition action* $\bar{a}_k$, *with* $\bar{\boldsymbol{\pi}} = (\bar{\pi}_k)_{k=1}^K$ *a joint policy. The semantics of* $\bar{M}$ *is given by all possible joint policies* $\Pi_{\bar{M}}$. *The partition* DecPOMDP *asks for a joint policy* $\bar{\boldsymbol{\pi}}^*$ *that maximises the expected utility as defined in Eq.* (1) *with* $\Pi_{\bar{M}}, \bar{\boldsymbol{\pi}}, \bar{\boldsymbol{a}}, \bar{\boldsymbol{o}}$ *as* $\Pi_M, \boldsymbol{\pi}, \boldsymbol{a}, \boldsymbol{o}$.

This formalism already bears benefits since it enables domain-specific compound actions or observations for partitions. It also allows for the same action and observation set to be available to each partition agent (Eq. (6)), i.e., $\bar{A}_k = A_k(X_k) = \{A_k(x_{k,1}), \ldots, A_k(x_{k,n_k})\}$ with $A_k(x_{k,i})$ the decision random variable for agent $i$ in partition $\mathfrak{I}_k$ (analogously for $\bar{O}_k$). If $\forall k : n_k = 1$, i.e., $N = K$, $\bar{M}$ is ground, i.e., $\bar{M} = M$.

The complexity results as given in Eqs. (4) and (5) for DecPOMDPs of Def. 1 translate directly to partitioned DecPOMDPs of Def. 2, referring to the range sizes of partition

actions and observations instead. Specifically, the respective function sizes $\bar{\mathbb{T}}$, $\bar{\mathbb{R}}$, and $\bar{\mathbb{O}}$ as well as the cost $\bar{\mathbb{C}}$ and policy space size $\bar{\bar{\mathbb{P}}}$ in a partitioned DecPOMDP lie in

$$\bar{\mathbb{T}} \in O\left(\bar{s}^2 \bar{a}^K\right) \quad \bar{\mathbb{R}} \in O\left(\bar{s}\bar{a}^K\right) \quad \bar{\mathbb{O}} \in O\left(\bar{s}\bar{o}^K\right) \quad (7)$$

$$\bar{\mathbb{C}} \in O\left(\bar{s}\bar{o}^{K\tau}\right) \quad \bar{\bar{\mathbb{P}}} \in O\left(\bar{a}^{\frac{K(\bar{o}^\tau - 1)}{\bar{o}-1}}\right) \quad (8)$$

with $\bar{s} = |ran(\bar{S})|$, $\bar{a} = \max_{k \in \{1,\ldots,K\}} |ran(\bar{A}_k)|$, and $\bar{o} = \max_{k \in \{1,\ldots,K\}} |ran(\bar{O}_k)|$. Assuming that $M$ and $\bar{M}$ model the same state space, then $s = \bar{s}$. With $N = K$, Eqs. (4) and (5) and Eqs. (7) and (8) coincide. With nonexistent or unused symmetries, the same holds even with Eq. (6).

Next, we specify two types of symmetries for $T$, $R$, and $\Omega$ to fulfil, specialise the partitioned DecPOMDP accordingly for a compact representation, and discuss their effect on the complexity results, starting with the more general symmetry.

## 3.2 COUNTING SYMMETRY

The basic assumption of lifting is indistinguishability between certain objects or individuals, or as in this paper, agents in DecPOMDPs. Next to the partitioning with Eq. (6), another consequence of the assumption is that it does not matter which agent performs a particular action within a partition, only how many (analogously for observations). E.g., if there are 10 agents in a partition of which five agents do one action and the rest do another action, then there exist $\binom{10!}{5!5!} = 252$ ways to have the 10 agents perform the 5 and 5 actions with the same outcome (multinomial coefficient, refer to Taghipour [2013] for details). The effect is that the transition, reward, and sensor functions map to identical values for these permutations of inputs within partitions. In the example, all of those 252 inputs map to some number $p$. We refer to this symmetry as *counting symmetry*.

Given counting symmetry, the question becomes how one can use it for a compact encoding in partitioned DecPOMDPs. In the example, it would be enough to store the number for each action, e.g., in a histogram of absolute numbers like $[5, 5]$, and map it to $p$. This type of intra-function symmetry has also been spotted in probabilistic inference and compactly encoded using histograms as range values of a so-called counting random variable (CRV, refer to Milch et al. [2008] for details). Formally, there are $r_k = |ran(A_k)|$ different possible actions per partition $\mathfrak{I}_k$, meaning, a partition action can be encoded using a histogram:

$$\{(a_l, n_l)\}_{l=1}^{r_k}, a_l \in ran(A_k), n_l \in \mathbb{N}^0, n_k = \sum_l n_l, \quad (9)$$

or $[n_1, \ldots, n_{r_k}]$ for short. CRVs bring us back to encoding identical action and observation sets with logical variables. Given a PRV such as $A_k(X_k)$, we count how often a particular range value is assigned to any grounding of $A_k(X_k)$, leading to a CRV $\#_{X_k}[A_k(X_k)]$, with histograms of Eq. (9) as a range (analogously for $O_k(X_k)$).

Hence, in the partitioned model $\bar{M}$ of Def. 2, $\bar{A}_k = \#_{X_k}[A_k(X_k)]$ and $\bar{O}_k = \#_{X_k}[O_k(X_k)]$ with the *domain* of $X_k$ being $dom(X_k) = \mathfrak{I}_k$. A joint action then turns into

$$\boldsymbol{a} = \{h_1, \ldots, h_K\}, h_k \in ran(\#_{X_k}[A_k(X_k)]).$$

The same holds for joint observations. The transition, reward, and sensor function, $T$, $R$, and $\Omega$ respectively, receive CRVs as inputs:

$$\bar{T}(\bar{S}', \bar{S}, \circ_{k=1}^K \#_{X_k}[A_k(X_k)]) \quad (10)$$

$$\bar{R}(\bar{S}, \circ_{k=1}^K \#_{X_k}[A_k(X_k)]) \quad (11)$$

$$\bar{\Omega}(\circ_{k=1}^K \#_{X_k}[O_k(X_k)], \bar{S}) \quad (12)$$

which allows for using a single mapping of a histogram to a number, e.g., $[5, 5] \mapsto p$, for permutations of inputs. We refer to this instantiation of a partitioned DecPOMDP (model $\bar{M}$) as a *counting DecPOMDP* (model $\bar{M}_c$).

The following results show that each counting model $\bar{M}_c$ has an equivalent ground model $M$ exhibiting counting symmetry, in which the same solutions are optimal. This leads us to the first result on agent tractability: A counting DecPOMDP depends on agent numbers exponentially in terms of the policy space but we achieve polynomial dependence in terms of representation size and cost.

**Lemma 1.** *A counting model $\bar{M}_c$ has an equivalent grounded model $M$ with counting symmetries, i.e., $gr(\bar{M}_c) = M$.*

*Proof.* Lifting is based on the grounding semantics [Sato, 1995], meaning an equivalence between lifted and ground operations. Thus, it also applies to CRVs. Grounding $\bar{M}_c$ means replacing each CRV $\#_{X_k}[A_k(X_k)]$ with decision random variables $\{A_k(x_{k,i})\}$, $i \in \mathfrak{I}_k$, analogously for observations, grounding the CRVs in the transition, reward, and sensor functions using so-called expansion [Taghipour, 2013], and setting $S$ to $\bar{S}$, yielding a grounded model. Because of expansion inverting the encoding of a counting symmetry in CRVs, $gr(\bar{M}_c)$ exhibits counting symmetries. $\square$

**Theorem 1.** *The optimal solution $\bar{\boldsymbol{\pi}}^*$ of a counting DecPOMDP in model $\bar{M}_c$ is also optimal in model $gr(\bar{M}_c)$.*

*Proof.* By Lemma 1, $\bar{M}_c$ is equivalent to $gr(\bar{M}_c)$. The difference when solving the problems on $\bar{M}_c$ and $gr(\bar{M}_c)$ with counting symmetry, respectively, lies in the policies searched. Namely, a search for $\bar{\boldsymbol{\pi}}^*$ in $\bar{M}_c$ skips those policies that have permutations of agents perform the same actions and perceive the same observations, which does not have an effect in the transition, reward, and sensor functions, and therefore have the same expected utility. Thus, the search for $\bar{\boldsymbol{\pi}}^*$ leads to a policy that is also optimal in $gr(\bar{M}_c)$. $\square$

**Theorem 2.** *A counting model allows for representation and cost to depend polynomially on $N$ (number of agents).*

*Proof.* The worst case complexities in terms of function sizes, evaluation cost, and policy space size of a counting DecPOMDP of model $\bar{M}_c$ with an equivalent ground DecPOMDP of model $M = gr(\bar{M}_c)$ lie in

$$\bar{\mathbb{T}}_c \in O\left(s^2 n^{Ka}\right) \ \bar{\mathbb{R}}_c \in O\left(sn^{Ka}\right) \ \bar{\mathbb{O}}_c \in O\left(sn^{Ko}\right) \ (13)$$

$$\bar{\mathbb{C}}_c \in O\left(sn^{K\tau o}\right) \quad \bar{\mathbb{P}}_c \in O\left(n^{a\frac{K(n^{\tau o}-1)}{n^o - 1}}\right) \quad (14)$$

with $s = |ran(S)|$, $n = \max_k n_k$, $a = \max_k |ran(A_k)|$, and $\bar{o} = \max_k |ran(O_k)|$, $k \in \{1, \ldots, K\}$. Because of the equivalence, $\bar{s} = s$. For $\bar{a}$ (and $\bar{o}$ analogously), it holds that $\bar{a} = \max_{k \in \{1,\ldots,K\}} |ran(\#_{X_k}[A_k(X_k)])|$. The range size of a CRV $\#_X[R(X)]$ with $|ran(R)| = r$ and $|dom(X)| = d$ is given by $\binom{d+r-1}{r-1} \leq d^r$ [Milch et al., 2008]. Therefore, $\bar{a}$ is replaced by $n^a$ and $\bar{o}$ by $n^o$ in Eqs. (7) and (8). As we assume $K \ll N$, $n$ is not much smaller than $N$. As such, the policy space still depends exponentially but representation size and cost only polynomially on $n < N$. $\square$

The $n$ represents the largest partition size, which is still large given $K$ is assumed to be small. An effect is that the naive solution of a brute-force search through the policies for an exact solution still depends exponentially on $n$ and therefore, the problem remains intractable even with counting symmetry. Solution techniques will have to do some further tricks, use heuristics, or stochastic methods to not depend on $n$ exponentially. Although the search space remains large here, it is not surprising that we cannot solve the combinatorial problem at once and it opens the door for interesting future work for a stochastic search in histogram spaces. However, there is also a positive side to this result as we do achieve a polynomial representation size even in the worst case.

Next, we consider isomorphic symmetry, which is actually able to render the problem tractable in agent numbers.

### 3.3 ISOMORPHIC SYMMETRY

This part starts with the model specification to see the origin of this symmetry and then infers the symmetry from it.

A natural next step in lifting is to consider PRVs as inputs instead of CRVs in the transition, reward, and sensor functions shown in Eqs. (10) to (12). The partitioned DecPOMDP and model is specialised with the PRV versions for actions and observations, i.e., $\bar{A}_k = A_k(X_k)$ and $\bar{O}_k = O_k(X_k)$ with $dom(X_k) = \mathfrak{I}_k$, yielding the following transition, reward, and sensor functions:

$$\bar{T}(\bar{S}', \bar{S}, \circ_{k=1}^K A_k(X_k))$$
$$\bar{R}(\bar{S}, \circ_{k=1}^K A_k(X_k))$$
$$\bar{\Omega}(\circ_{k=1}^K O_k(X_k), \bar{S})$$

We call this form of partitioned DecPOMDP (model $\bar{M}$) an *isomorphic DecPOMDP* (model $\bar{M}_i$).

The grounding semantics also applies: $\bar{T}$, $\bar{R}$, and $\bar{\Omega}$ turn into combinations over all groundings $\boldsymbol{D} = \times_{k=1}^K dom(X_k)$, each grounding with identical mappings as of $\bar{T}$, $\bar{R}$, and $\bar{\Omega}$. E.g., the result for $\bar{T}$ is $\bar{T}(\bar{S}', \bar{S}, \circ_{k=1}^K A_k(x_{k,i_k}))$ for all groundings $(x_{1,i_1}, \ldots, x_{K,i_K}) \in \boldsymbol{D}$, which are combined using multiplication. Rewards are additive, i.e., combined using summation, which implies a preferential independence [Russell and Norvig, 2020] over which agent incurs a reward, which is reasonable under indistinguishability.

A joint distribution factorising into such identical functions is the main symmetry used in LVE, encoded using logical variables in PRVs. Grounded out, subgraphs coming from the same parameterised function are isomorphic. Therefore, we refer to this symmetry as *isomorphic symmetry*. Isomorphic models translate into counting models, which we show next, meaning Theorem 1 also holds in this case.

**Lemma 2.** *An isomorphic model $\bar{M}_i$ has an equivalent counting model $\bar{M}_c$.*

*Proof.* A model with isomorphic symmetry has a corresponding isomorphic DecPOMDP based on the grounding semantics with PRVs as inputs in the functions $\bar{T}$, $\bar{R}$, and $\bar{\Omega}$. All PRVs are count-convertible as each contains only one logvar that does not occur in any other PRV and the domains are mutually exclusive, automatically fulfilling the preconditions of count-conversion.[2] For $\bar{T}$ and $\bar{\Omega}$, standard count-conversion can be used [Taghipour, 2013]. Because of $\bar{R}$ having additive semantics, an additive but otherwise unchanged count-conversion is necessary; see the supplementary material for a formal definition. The count-conversions result in functions of Eqs. (10) to (12), i.e., a counting model. $\square$

**Corollary 2.1.** *If a model $M$ exhibits isomorphic symmetry, it also exhibits counting symmetry.*

*Proof.* By the grounding semantics, such a model $M$ can be encoded in an isomorphic model $\bar{M}_i$, which has an equivalent counting model by Lemma 2, which has a ground model with counting symmetry by Lemma 1. $\square$

The reverse is not necessarily true: One can build a counter example by turning an isomorphic model into a counting one and then changing the mapped value of one input sequence, destroying the factorisation. Thus, isomorphic symmetry is a stricter symmetry but it allows for cutting down the policy space. The reason lies in the PRV version implying a form of independence on the agent level. Each agent within a partition operates independent of other agents in the partition. As the agents of a partition are indistinguishable, each agent has the same optimal local policy in this setting. Thus, one only needs to consider a representative agent and search through

---

[2]Isomorphic models fall into a specific class of liftable models with PRVs with at most one logical variable [Taghipour, 2013]

its possible actions and observations of $A_k$ and $O_k$ to find its optimal policy, which also applies to all other agents of the partition. Therefore, we can trim the search space to cover only $ran(A_k)$ for isomorphic DecPOMDPs instead of $ran(\#_{X_k}[A_k(X_k)])$ for equivalent counting DecPOMDPs. Formally, we have the following lemma.

**Lemma 3.** *In an isomorphic DecPOMDP with model $\bar{M}_i$, partition policies are defined over $ran(A_k)$ and $ran(O_k)$.*

*Proof sketch.* The model $\bar{M}_i$ is basically a parameterised probabilistic sequential model with three parametric factors, i.e., factors with PRVs as inputs, namely, $\bar{T}_i$, $\bar{R}_i$, and $\bar{\Omega}_i$. The inputs are time-stamped by $t$, i.e., $S_t$, $\bar{A}_t$, and $\bar{O}_t$. A policy prescribes an action $a$ to perform for each possible observation history $o_{0:\tau}$ of length $\tau$ for each agent. To determine a joint action $\boldsymbol{a}$ for a joint observation history $\boldsymbol{o}_{0:\tau}$, $\boldsymbol{o}_{0:\tau}$ becomes evidence in $\bar{M}_i$. Since there are only a bounded number of histories possible in a partition, the agents in a partition can be partitioned again based on the histories. That is $\boldsymbol{o}_{0:\tau}$ is turned into lifted evidence and the partitions are split on the evidence using the split operator [Taghipour, 2013]. So, each of the $K$ partitions has its own logical variable $X_{k,j}$ with a domain that is disjoint from all other domains and the same evidence for the agents of $X_{k,j}$.

Next, we unroll the model for $\tau$ time steps, i.e., instantiate the split functions by replacing $t$ with each value in $\{0, \ldots, \tau\}$, to handle the evidence over $\tau$ time steps. Then, the model absorbs the evidence in each sub-partition, eliminating $O_k(X)$ from each sub-partition and each time step, using the lifted absorption operator [Taghipour, 2013]. For the different time steps, we now have $\bar{T}_i(\bar{S}', \bar{S}, \circ_{k=1}^{K} \circ_j A_k(X_j))$, $\bar{R}_i(\bar{S}, \circ_{k=1}^{K} \circ_j A_k(X_j))$, and $\bar{\Omega}_i(S)$ where $j$ iterates over all existing sub-partitions of a partition.

Next, we follow the semantics of parameterised models to show that considering only $ran(A_k)$ is correct. The semantics prescribe the following: (i) Ground the model. (ii) Join all instantiated factors into one large factor. (iii) Sum out the state variables $S_t$, $S_t'$ for all instantiated $t$. (iv) Pick the MEU actions for the different observation histories. The task in terms of the proof is to show that the same $\arg\max$ actions are chosen for agents from the same sub-partition.

Step (i) makes every constant covered by a PRV explicit by having each constant appear explicitly in a grounding (in contrast to the $X$, which hides the constants). Making the constants explicit, i.e., no longer hiding them behind an $X$, can also be achieved by turning each PRV into a CRV, where each constant appears in the counts of the histograms, after which no constants are hidden anymore. So, we use count-converting instead of grounding in (i) for the purpose of this proof. (Count-conversions are applicable as the domains are disjoint with no inequalities between logical variables and no PRVs or logical variables recurring in the same function.) Then, we need to show that peak-shaped histograms (in

Eq. (9), $n_l = n_k$, for $l' \neq l : n_{l'} = 0$) are chosen as $\arg\max$ actions, which means that agents of the same sub-partition perform the same action.

To show that only peak-shaped histograms are relevant, we look at the count-conversions of the first step. Consider a minimum example of a function $\phi_i(S, A(X))$ and its ground and count-converted versions $\phi$ and $\phi_c$ with $|dom(X)| = 2$:

| $S$ | $A(X)$ | $\phi_i$ |
|---|---|---|
| $s^0$ | $a^0$ | $p_1$ |
| $s^0$ | $a^1$ | $p_2$ |
| $s^1$ | $a^0$ | $p_3$ |
| $s^1$ | $a^1$ | $p_4$ |

| $S$ | $A(x_1)$ | $A(x_2)$ | $\phi$ |
|---|---|---|---|
| $s^0$ | $a^0$ | $a^0$ | $p_1^2$ |
| $s^0$ | $a^0$ | $a^1$ | $p_1 \cdot p_2$ |
| $s^0$ | $a^1$ | $a^0$ | $p_2 \cdot p_1$ |
| $s^0$ | $a^1$ | $a^1$ | $p_2^2$ |
| $s^1$ | $a^0$ | $a^0$ | $p_3^2$ |
| $s^1$ | $a^0$ | $a^1$ | $p_3 \cdot p_4$ |
| $s^1$ | $a^1$ | $a^0$ | $p_4 \cdot p_3$ |
| $s^1$ | $a^1$ | $a^1$ | $p_4^2$ |

| $S$ | $\#_X[A(X)]$ | $\phi_c$ |
|---|---|---|
| $s^0$ | $[0, 2]$ | $p_2^0 \cdot p_1^2$ |
| $s^0$ | $[1, 1]$ | $p_2^1 \cdot p_1^1$ |
| $s^0$ | $[2, 0]$ | $p_2^2 \cdot p_1^0$ |
| $s^1$ | $[0, 2]$ | $p_4^0 \cdot p_3^2$ |
| $s^1$ | $[1, 1]$ | $p_4^1 \cdot p_3^1$ |
| $s^1$ | $[2, 0]$ | $p_4^2 \cdot p_3^0$ |

The versions $\phi$ and $\phi_c$ show exactly how a CRV is simply another encoding of the same information stored in a grounded version. It also highlights that an isomorphic symmetry is always also a counting symmetry. The important part for the proof is that whatever the actual values of the different $p$'s are, when the PRVs are count-converted (or grounded), the maximum value for a given state $s$ will always occur where the exponent $e$ is largest as one of the $p$'s will be the largest and it will bring the most to take this $p$ to the power of the largest number possible. And the largest exponent possible occurs in a peak-shaped histogram where one position takes all the available elements. With the additive semantics of rewards, the rewards are added up (and not multiplied) with the exponent being a factor but the argument is the same: the maximum value occurs where the largest $p$ meets the largest $e$, which occurs in a peak-shaped histogram.

Non-peak-shaped histograms can only catch up to peak-shaped histograms, namely, if $p$'s are equal. Then, the solution will not be unique. As we are not interested in all solutions, focusing on peak-shaped histograms is still correct.

So, we have that one of the inputs with a peak-shaped histogram maps to the largest value, i.e., a peak-shaped histogram is the $\arg\max$ action at the moment. In the functions of a DecPOMDP model, we have $K$ PRVs to count-convert. However, the result still remains the same: For each of the count-conversions individually, it holds with the arguments above that the $\arg\max$ action is one of the peak-shaped his-

tograms. Since the count-conversions are applied iteratively, each CRV still has the maximum value where a peak-shaped histogram occurs, and these maximum values occur for those inputs in which the previous count-conversion had maximum values, which were also peak-shaped. So, after count-converting every logical variable in $\bar{T}_i$ and $\bar{R}_i$ of each sub-partition ($\bar{\Omega}_i$ does not contain any logical variable as we eliminated them using the observation histories as evidence), the lines with the largest values will be those where peak-shaped histograms occur together.

Step (ii) says to join all functions. The largest values occur for each possible state where peak-shaped histograms meet as they bring with them the largest current $p$.

Step (iii) requires summing out all non-decision CRVs, i.e., all state variables over the different time steps. Summing out adds up values that occur for the different states given the same input sequence of the CRVs, which again does not change any $\arg\max$ actions as non-peak-shaped inputs cannot catch up with those peak-shaped inputs, where the largest values reside and are now added up.

Step (iv) is already the decision on the $\arg\max$ action, where we pick the input mapping to the largest value, which has peak-shaped histograms by the arguments above. Peak-shaped histograms prescribe the same action for all agents in a sub-partition, meaning it is enough to consider $ran(A_k)$ in each sub-partition.

The remaining part of the proof has to show that it is also enough to consider $ran(O_k)$ and to not consider each possible sub-partition as we have done so far. Each sub-partition has peak-shaped histograms as $\arg\max$ actions. An effect is that the $\arg\max$ actions do not depend on partition sizes. Only the size of the peak changes. Therefore, for all the other possible observation histories $o'_{0:\tau}$, we again get sub-partitions that have observation histories as evidence like with $o_{0:\tau}$ but with different sub-partition sizes. However, the result in terms of the $\arg\max$ actions per sub-partition will be the same. Therefore, it is enough to consider what happens for one representative agent for each possible observation history of that agent as looking at $n_{k,j}$ agents in a sub-partition does not change the outcome compared to looking at one agent. That is, considering only $ran(O_k)$ and the histories possible with $ran(O_k)$ is sufficient. This concludes our proof. □

**Corollary 3.1.** *Partition sizes do not influence a decision, but only the overall expected utility in isomorphic models.*

The proof of Lemma 3 only works because the CRVs come from PRVs. If one would have a CRV in the original model, a non-peak-shaped histogram could get such a high reward that it will outweigh all others and be the $\arg\max$ action. Corollary 3.1 follows directly from the proof. Thus, one can ignore partition sizes during policy evaluation if only interested in a ranking. Lemma 3 brings us to our second main

result, which is an overwhelmingly positive one: Isomorphic DecPOMDPs allow for agent tractability and a complete independence of agent numbers if one ranks policies.

**Theorem 3.** *An isomorphic model leads to* tractability *regarding the number of agents $N$.*

*Proof.* Complexity-wise, Lemma 3 has the effect that $\bar{a} = \max_k |ran(A_k)|$ and $\bar{o} = \max_k |ran(O_k)|$, $k \in \{1, \ldots, K\}$ in Eqs. (7) and (8), which is equal to the sizes $a$ and $o$ of Eqs. (4) and (5). Thus, the sizes $\bar{\mathbb{T}}_i$, $\bar{\mathbb{R}}_i$, and $\bar{\mathbb{O}}_i$, the evaluation cost $\bar{\mathbb{C}}_i$, and the policy space size $\bar{\mathbb{P}}_i$ lie in:

$$\bar{\mathbb{T}}_i \in O\left(s^2 a^K\right) \quad \bar{\mathbb{R}}_i \in O\left(s a^K\right) \quad \bar{\mathbb{O}}_i \in O\left(s o^K\right) \quad (15)$$

$$\bar{\mathbb{C}}_i \in O\left(\log_2(n) s o^{K\tau}\right) \quad \bar{\mathbb{P}}_i \in O\left(a^{\frac{K(o^\tau - 1)}{o-1}}\right) \quad (16)$$

with $s = |ran(S)| \,(= \bar{s})$ and $n = \max_k n_k$. The $\log n$ for $\bar{\mathbb{C}}_i$ appears when including partition sizes into the evaluation (they occur as exponents and exponentiation has a complexity of $\log_2 n$). Then, the naive solution of a brute-force search through the policy space no longer depends on $N$ exponentially, making the problem tractable for models with isomorphic symmetries regarding agent numbers. □

## 3.4 DISCUSSION

This part discusses expressiveness, partition-level symmetries and independence, identifying symmetries, and the connection to LVE, with a concluding note on going from joint policy to agents in symmetric DecPOMDPs.

**Expressiveness** Models with counting symmetry allow for encoding outcomes based on proportions of agents acting or observing. And even though the worst case shows an exponential dependence on the number of agents $N$, domain-specific knowledge may help to prune the search space as only certain limits may apply (e.g., at least 30,000 actions of a certain type) or histograms in increments of 1 are not sensible (cf. $[0, 50.000]$ and $[1, 49.999]$).

Isomorphic models cannot express proportions, which are not always necessary. If each agent's action only has local effects (consider navigating on a grid), the model may be isomorphic. A joint reward may only depend on $S$, i.e., $R(S)$, which is independent of any action and therefore automatically isomorphic. If one can assume that partition agents work in close proximity or in a comparable setup, observations may often be identical, even if noisy, opening up future work on approximating observations.

Consider the well-known DecTiger benchmark [Nair et al., 2003] as an example. Please find a full specification and discussion in the supplementary material. Agents have the same set of actions and observations available. The functions $T$, $R$, and $\Omega$ exhibit a counting symmetry, whereas only $\Omega$ is also isomorphic. $T$ and $R$ do not factorise accordingly. In

case of $T$, the reason lies in the automatic reset built into it. For $R$, one could adjust its mapping of four histograms to achieve a factorisation. However, the adjusted $R$ would not be able to capture that both agents agreeing on an action even though it opens the door to the tiger costs them less ($-50$) than opening different doors ($-100$). Here, one would need counting. Nonetheless, an isomorphic model would exclude any policy where one agent opens a door while the other agent either listens or opens the other door, which are not optimal (and in a sense stupid) action combinations.

Let us briefly consider the nanoscale medical system for robust diagnosis with, e.g., four types of agents. Consider two possible actions, releasing or not releasing a load, and one possible observations of detecting a virus signature. In such a setting, we have four partitions, each with the same actions and observations. Preliminary experiments have shown that each partition may have around 64,000 agents, making the agent set at least of size $4 \cdot 64{,}000$, which is not computable on the ground level.

The question then becomes how to model the transition, reward, and observation functions, i.e., how much does one approximate. Counting symmetries would be the more appropriate choice if one wants to model that a certain fraction of agents has to perform an action for a reward. Braun et al. [2021] provide a more detailed description of a nanoscale medical system, especially regarding counting symmetries. In terms of specifying the model, the required memory would be only polynomial in size. However, solving such a problem with histograms that represent $4 \cdot 64{,}000$ agents is also not computable because of the large search space for policies. With such large agent numbers, we want to assume isomorphic symmetries, which would involve that each agent acts independently of the other agents in their environment. Such an assumption is very approximate at first glance but since nanoagents have only limited online computation power and memory available, approximating them as acting independently of all other agents appears reasonable. Therefore, the goal would be to set up an isomorphic model.

**Identifying Symmetries**  To use symmetries, one has to identify them in the problem at hand. In case of the nanoscale medical system, symmetries lie in the nature of the problem as there are a handful of types of agents but thousands of agents of each type and they can be directly used when specifying the problem instance. If one does have a full propositional model, then one has to check for the symmetries in the probabilities, i.e., values reoccurring, of the functions after one has checked that there are indeed groups of agents that have the same sets of actions and observations available. A first way would be to start with checking if Eq. (6) holds and then looking at the numbers in transition, reward, and sensor functions for symmetries. The colour passing algorithm by Ahmadi et al. [2013] is a good starting point for automating the search for symmetries.

The idea is of course to learn a counting or isomorphic model directly to capitalise on the fact that these models are smaller and require fewer parameters to learn. Like in factored PGMs, one does not want to learn the full joint and then test for independences to get the factorisation and then for symmetries to lift the model.

**Partition-level Symmetry: Mixed Symmetries**  The analysis above focuses on complete models exhibiting one type of symmetry for ease of exposition. However, models can of course display a mix of the two symmetries with certain partitions exhibiting counting symmetry and others isomorphic symmetry, which leads to the following definition.

**Definition 3.** *A model $M$ is* symmetric *if it can be partitioned with Eq. (6) and $T$, $R$ and $\Omega$ show one of the following symmetries for each of the resulting partitions:*

*(1)* Counting*: $T$, $R$ and $\Omega$ map to the same values for permutations of actions and observations in a partition.*

*(2)* Isomorphic*: $T$, $R$ and $\Omega$ factorise into a set of identical functions with representative agents.*

On the modelling side, a partitioned DecPOMDP would then consist of PRVs and CRVs for actions and observations and as inputs to the transition, reward, and sensor functions, which would compactly encode the corresponding behaviour depending to the symmetry for each partition. Of course, the assumptions about independence and symmetry have to hold between the different partitions for mixed symmetries to be able to occur, which may not be realistic for real-world data. Weakening the assumptions or approximating the symmetries in a bounded way lies ahead in future work.

In terms of complexity, we can distinguish between $I$ isomorphic and $C$ counting partitions ($I + C = K$):

$$\bar{\mathbb{T}} \in O\left(s^2 a^I n^{Ca}\right) \quad \bar{\mathbb{R}} \in O\left(sa^I n^{Ca}\right) \quad \bar{\mathbb{O}} \in O\left(so^I n^{Co}\right)$$

$$\bar{\mathbb{C}} \in O\left(so^I \tau n^{C\tau o}\right) \quad \bar{\mathbb{P}} \in O\left(a^{\frac{I(o^\tau - 1)}{o - 1}} n^{a\frac{C(n^{\tau o} - 1)}{n^o - 1}}\right)$$

with $s = |ran(S)|$, $n = \max_k n_k$, $a = \max_k |ran(A_k)|$, and $\bar{o} = \max_k |ran(O_k)|$, $k \in \{1, \ldots, K\}$. Partitions of size 1 belong to isomorphic partitions, i.e., count towards $I$.

**Partition-level Independence**  It might be reasonable to assume that partitions are only connected through the joint state and therefore act and observe independently from each other. The consequence for the transition, reward, and sensor function $T$, $R$, and $\Omega$ of a symmetric model is that they factorise into sets of $K$ functions $T_k$, $R_k$, and $\Omega_k$:

$$\bar{T}_k(\bar{S}', \bar{S}, \bar{A}_k) \quad \bar{R}_k(S, \bar{A}_k) \quad \bar{\Omega}_k(\bar{O}_k, S)$$

with $\bar{A}_k = \#_{X_k}[A_k(X_k)]$ and $\bar{O}_k = \#_{X_k}[O_k(X_k)]$ in the counting case and $\bar{A}_k = A_k(X_k)$ and $\bar{O}_k = O_k(X_k)$ with $n_k$ as exponent in the isomorphic case. The effect on the

complexities in both settings is that the exponent $K$ becomes a regular factor. The effect is not that large as $K$ is assumed to be small. Nonetheless, this setting presents the best case with the exponent $N$ reduced to a factor $K$ in an isomorphic DecPOMDP. It also means that we move towards a POSG as each partition has its own reward function $\bar{R}_k$.

**Connection to LVE** One can draw parallels between counting and isomorphic models on the DecPOMDP side and count-conversion and lifted summing out on the LVE side, respectively. In terms of complexity, lifted summing out also has a better bound than that of count-conversion. The term that characterises the effort is the lifted width, which basically bounds the largest possible intermediate result during inference [Taghipour, 2013]. Without any CRVs, i.e., no CRVs in the model and no count-conversions in inference, the size in $n$, the maximum domain size, is bounded by $\log_2 n$, also coming from exponentiation in inference. With CRVs, it is polynomial in $n$. Here, the problem indeed becomes tractable for both counting and isomorphism regarding domain sizes in contrast to DecPOMDPs where the counting case remains intractable because of the policy space and only the isomorphic case becomes tractable.

**Isomorphic Models, DecPOMDP Solvers, & New Query Types** Given an isomorphic model, Lemma 3 allows for adapting any solution technique for a (ground) DecPOMDP like the Joint Equilibrium Search for Policies by Nair et al. [2003] by using it to find an optimal response policy for a representative and incorporating partition sizes into the expected utility calculation if necessary, which leads to a Nash equilibrium policy. Empirical evaluations of ground techniques usually work with single-digit agent numbers, which is the assumed order of $K$. Since we can transfer the algorithms to the isomorphic setting, the results still hold but now for number of partitions if one ranks policies. As such, we can now let the number of agents increase significantly in isomorphic models.

In addition, this new setting allows for a new query type apart from a joint policy based on partitions, paraphrased as: "*How many agents do I need to reach a minimum?*" In scenarios, where we are interested in reaching a minimum expected utility $U_{min}$ by possibly investing more agents, we can use any algorithm to solve an isomorphic DecPOMDP instance in model $\bar{M}_i$ as argued above with given partition sizes, yielding a joint policy $\bar{\pi}^*$ with an optimal expected utility $U_{\bar{M}_i}(\bar{\pi}^*)$. Then, since the agent numbers are still a part of the model, one can define an optimisation problem around this where we are interested in the number of agents needed per partition to reach $U_{min} > U_{\bar{M}_i}(\bar{\pi}^*)$. Since we have an isomorphic DecPOMDP, having more agents in a partition does not have any intra-partition effects because of the quasi-independence between agents and therefore does not lead to changes in $\bar{\pi}^*$. However, having more agents does influence the $U$ value of the joint policy.

**From Policies to Functional Agents** When implementing a multi-agent system, the agents can be set up (or built, in the nanoscale system setting) according to the joint policies: In the isomorphic case, all agents in a partition behave according to the same policy and can be set up with that policy. In the counting case, the counting policy can be grounded, leading to policies for individual agents. Grounding is always possible with lifted constructs, which would require the so-called expansion operator by Taghipour [2013] for grounding CRVs. The agents can then be set up according to the (grounded) local policies. If combining (counting) all those local policies, a policy emerges equivalent to the original counting policy.

In summary, symmetric DecPOMDPs allow for partitions and a compact representation of actions, observations, and policy. In addition, the isomorphic variant allows for using existing algorithms with large agent sets in partitions and solving new queries for partition sizes.

## 4 CONCLUSION & FUTURE WORK

Inspired by the symmetries of counting and isomorphism in lifting, this paper analyses the effect of symmetries on DecPOMDPs. To this end, it specifies what counting and isomorphic symmetry look like in ground models, allowing for a compact encoding using a partitioned DecPOMDP, which enables an explicit representation of partitions in the problem specification and the solution. The formal analysis shows that the symmetries allow for reducing the representation complexity and the cost of policy evaluation dependency to polynomial. There still is an exponential dependence on the agent numbers in the policy space for the counting case, which is unsurprising given the combinatorial explosion and the symmetry used. In the case of isomorphism, however, the problem becomes tractable regarding agent numbers and even allows for reusing existing ground solution methods while also enabling new query types.

Future work focuses on stochastic search for symmetric DecPOMDPs, especially counting DecPOMDPs. Other interesting avenues include continuous state and observation considerations, communication as well as combining lifting for agents with lifting on the state space as in FOMDPs.

### Acknowledgements

The research of Marcel Gehrke was funded by the Deutsche Forschungsgemeinschaft (DFG, German Research Foundation) under Germany's Excellence Strategy – EXC 2176 'Understanding Written Artefacts: Material, Interaction and Transmission in Manuscript Cultures', project no. 390893796. The research was conducted within the scope of the Centre for the Study of Manuscript Cultures (CSMC) at Universität Hamburg.

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
