# OpenReview forum: "Lifting in Multi-agent Systems under Uncertainty"
_auai.org/UAI/2022/Conference — UAI 2022 Poster_

### Official Review · Reviewer_2U49 · 2022-03-29

**Q2(1) Originality/Novelty:** 3
**Q2(2) Significance/Impact:** 3
**Q2(3) Correctness/Technical Quality:** 2
**Q2(6) Clarity Of Writing:** 3
**Q6 Overall Score:** 7
**Q8 Confidence In Your Score:** 2

**Q1 Summary And Contributions:**

The paper proposes the exploitation of agent symmetries in decentralized partially observable Markov decision process (DecPOMDP), including counting and isomorphic symmetries, for more tractable solutions of optimal policies.

**Q2 Assessment Of The Paper:**

More detailed information regarding each of these aspects is given below:

**Q2(5) Reproducibility:**

2: Fair: Key resources (e.g., proofs, code, data) are unavailable but key details (e.g., proof sketches, experimental setup) are sufficiently well-described for an expert to confidently reproduce the main results.

**Q3 Main Strengths:**

The idea of exploiting agent symmetries to overcome the intractability of solving DecPOMDP seems novel and can improve the efficiency of the optimization process significantly.

**Q4 Main Weakness:**

The presentation of the paper is not always good and sometimes lacks mathematical precision (See the issues to be clarified below). The application scope of the proposed methodology is also somewhat unclear.

**Q5 Detailed Comments To The Authors:**

While the paper presents valuable contribution to overcome intractability of solving  DecPOMDP,  its presentation is sometimes obscure. To improve the readability, the following issues should be clarified.

First, it seems that symmetries within the agent set depend on the nature of the problem to be modeled. Hence, for the proposed method to be practically useful,  there is a prerequisite of recognizing the existence of  indistinguishability between agents. However, in the paper, this is not mentioned at all. Is it easy for modelers to recognize counting or isomorphic symmetries in a given DecPOMDP? Or are there any guidelines for them to exploit agent symmetries? More specifically, it is only mentioned that (6) is a necessary condition for partitioning the agent set, but is it also a sufficient condition? Otherwise, is there an algorithmic way to decide whether and  what kind of symmetries exist in a given model M (e.g. to determine how to partition the agent set and to check whether transition, reward, and sensor functions satisfy counting symmetries)? If yes, is there extra cost for such kinds of algorithms in terms of complexity? In other words, when considering the complexity of the proposed method, do we also have to add the complexity of transforming a model into its counting or isomorphic version?

Second, in the partitioned model, a policy is defined as partition observation (histories) to partition actions. It is somewhat imprecise what partition actions or observations mean. By instantiating it to counting or isomorphic models, this seemingly means the histogram of actions (resp. observations) in a group of agents or representative ones in that group. Then, there is the problem of intra-group communication cost. When an agent set is partitioned into groups, it is unclear whether each agent in a group still acts independently or the whole group act collectively as one? If it is the former, then it seems that each agent in a group must also know observations of other agents in the same group for the partition policy to work; it it is the latter, then it seems that there must be a centralized agent in each group to collect all observations of agents in the group to make the decision. Both cases will incur communication costs between agents in the same group.  It seems that this kind of cost cannot be ignored and should be taken into account for the complexity analysis of the proposed method.

Finally, the notation \circ in Eqs. (10)-(12) is not explained explicitly. Does it mean a concatenation or Cartesian product or anything else?


**Q7 Justification For Your Score:**

The work itself seems valuable, although the presentation could be improved.

**Q9 Complying With Reviewing Instructions:**

1: Yes.

---

### Official Review · Reviewer_qW85 · 2022-04-09

**Q2(1) Originality/Novelty:** 3
**Q2(2) Significance/Impact:** 2
**Q2(3) Correctness/Technical Quality:** 4
**Q2(6) Clarity Of Writing:** 3
**Q6 Overall Score:** 6
**Q8 Confidence In Your Score:** 4

**Q1 Summary And Contributions:**

The paper studies sub-classes of DEC-POMDPs where symmetries can be exploited to reduce the complexity and improve the tractability. The authors first introduce partitioned Dec-POMDPs and then study two kinds of symmetries: counting symmetries where the outcome of an action only relies on the number of agents executing an action, and isomorphic symmetries where the transition, the observation and the reward functions can be factorized into identical functions per agent.

**Q2 Assessment Of The Paper:**

More detailed information regarding each of these aspects is given below:

**Q2(5) Reproducibility:**

3: Good: Key resources (e.g., proofs, code, data) are available and key details (e.g., proofs, experimental setup) are sufficiently well-described for competent researchers to confidently reproduce the main results.

**Q3 Main Strengths:**

The paper presents new directions to improve the scalability of DEC-POMDPs and should contribute to advance the state of the art.
The paper is technically sound.


**Q4 Main Weakness:**



The paper focuses on identifying sub-classes of DEC-POMDPs where symmetries make the problem more tractable. However, the authors neither study algorithms to solve such problems nor investigate the effect of exploiting these symmetries in real-world problems. In many cases, the symmetries may not reduce enough the representation when the number of partitions k is large.  It is thus difficult to assess the effective gain in tractability of the approach.
Moreover, the relevance of such sub-classes of problems remains difficult to estimate. Real-world problems exhibiting such symmetries seem to be very rare. When the performance of the agents rely on cooperative interactions, it seems difficult to use such sub-problems. In the dec-tiger problem, the symmetric models fail to capture the cooperation for opening the doors.

**Q5 Detailed Comments To The Authors:**

 In the dec-pomdp benchmark (available with madp), which problems can be formalized by symmetric models? It would also be interesting to illustrate the relevance of each kind of symmetry on concrete multiagent cooperative decision problems.

The writing on the paper is quite austere. It is difficult to assess the meaning of each symmetry. Providing problem examples(defined as a DEC-POMDP) for which each symmetry applies would improved the readability of the paper.  The Dec-Tiger translation provided in Appendix is useful but fails to exemplify the relevance of the isomorphic symmetry.

In the background section, it would be interesting to relate this approach to DEC-POMDPs with observation and transition independence. Indeed, this former work is in the same vein as the one of the paper.

Note that JESP algorithm is not guarantee to return an optimal solution, it returns a Nash equilibrium.


**Q7 Justification For Your Score:**


The paper tackles new directions to improve the tractability of dec-pomdps.
However, the relevance of the symmetries to reduce the complexity in real world problems is difficult to assess. Indeed, these symmetries are likely to apply in very restricted domains.

**Q9 Complying With Reviewing Instructions:**

1: Yes.

---

### Official Review · Reviewer_SiEE · 2022-04-13

**Q2(1) Originality/Novelty:** 1
**Q2(2) Significance/Impact:** 2
**Q2(3) Correctness/Technical Quality:** 3
**Q2(6) Clarity Of Writing:** 2
**Q6 Overall Score:** 6
**Q8 Confidence In Your Score:** 4

**Q1 Summary And Contributions:**

The paper describes theoretical properties of lifted decentralised partially observable Markov decision problems (DecPOMDP), which formalize collaborative multi-agent decision making in the present of symmetries among the multiple agents.


**Q2 Assessment Of The Paper:**

More detailed information regarding each of these aspects is given below:

**Q2(5) Reproducibility:**

2: Fair: Key resources (e.g., proofs, code, data) are unavailable but key details (e.g., proof sketches, experimental setup) are sufficiently well-described for an expert to confidently reproduce the main results.

**Q3 Main Strengths:**

The paper defines two types of symmetries takes from the lifted inference literature: counting and isomorphic. The former encodes the notion that *which* agents perform certain actions does not matter, only *how many* of them. The latter encodes the situation in which agents are completely independent.

The main results of the paper are that both symmetries allow representation size and cost independent on the number of agents whereas isomorphic symmetry allow even policy size and re-use of solvers for the ground (non-lifted/non-symmetric) case.

**Q4 Main Weakness:**

The main explanations of the paper are in English and kept at a very abstract level, which sometimes makes it hard to follow. There are many examples, one of which is the sentence "Step (i) makes every constant covered by a PRV explicit, which is also achieved by turning each PRV into a CRV.". In this case, it is not so clear what "explicit" means and how this is preserved by a conversion into a CRV.

The results are somewhat intuitively expected, but it is still good to have them investigated and recorded.

**Q5 Detailed Comments To The Authors:**

I don't have many detailed comments other than suggest a more mathematical description of the results (which I appreciate may not be easy to achieve).

I found the explanation of mixed symmetries somewhat incomplete. I don't see how taking separating partitions into those with counting symmetry and those with isomorphic symmetry is enough, because it is unclear why they will not interact in some way that renders that separation ineffective.

**Q7 Justification For Your Score:**

The paper describes results that are somewhat expected but still interesting. It is hard to follow but it is correct as far as I can tell.

**Q9 Complying With Reviewing Instructions:**

1: Yes.

---

### Official Review · Reviewer_84C7 · 2022-05-04

**Q2(1) Originality/Novelty:** 3
**Q2(2) Significance/Impact:** 3
**Q2(3) Correctness/Technical Quality:** 3
**Q2(6) Clarity Of Writing:** 3
**Q6 Overall Score:** 6
**Q8 Confidence In Your Score:** 4

**Q1 Summary And Contributions:**

This paper presents a Relational POMDP model together with symmetry assumptions/constraints and shows that the principle of lifting from Relational Probabilistic Inference can be applied to this model as well. It provides theorems and proofs showing that the relational model and its solution are equivalent to a ground model and its solution.

**Q2 Assessment Of The Paper:**

More detailed information regarding each of these aspects is given below:

**Q2(4) Quality Of Experiments (Optional):**

1: Poor: The experimental evaluation is flawed or the results fail to adequately support the main claims.

**Q2(5) Reproducibility:**

3: Good: Key resources (e.g., proofs, code, data) are available and key details (e.g., proofs, experimental setup) are sufficiently well-described for competent researchers to confidently reproduce the main results.

**Q3 Main Strengths:**

Theoretical results are shown and the model is explained clearly.

**Q4 Main Weakness:**

Experimental results and real-world applications are missing.


**Q5 Detailed Comments To The Authors:**

This paper presents a Relational POMDP model together with symmetry assumptions/constraints and shows that the principle of lifting from Relational Probabilistic Inference can be applied to this model as well. It provides theorems and proofs showing that the relational model and its solution are equivalent to a ground model and its solution.

Generalizations of the first model and its solution methods are shown (Partition-Level symmetry and mixing different symmetries) with results that were shown previously in the literature in the context of inference ([Braz et al 2005]).

It would be helpful to examine and analyze the results of such interactions in real models pertaining to real-world situations (e.g. flying drones forming different shapes, or psychological models of peer-to-peer interactions).

There are no experimental results, and little motivation is given in terms of real world examples that exhibit the symmetries described here.
It would help this paper greatly to have such.


**Q7 Justification For Your Score:**

The paper provides a theoretical advance that is useful, but the impact is likely to be small due to the little attention to real-world applications or experimental results.


**Q9 Complying With Reviewing Instructions:**

1: Yes.

---

### Decision · Program_Chairs · 2022-05-15

**Decision:**

Accept (Poster)

**Comment:**

Meta Review: Congratulations. The reviewers agreed that this was a paper worth of acceptance. Please improve the paper as outlined in your responses.